# Exciton fission in monolayer transition metal dichalcogenide semiconductors

A. Steinhoff[1], M. Florian[1], M. Rösner[1,2,4], G. Schönhoff[1,2], T.O. Wehling[1,2,3] & F. Jahnke[1,3]

When electron-hole pairs are excited in a semiconductor, it is a priori not clear if they form a plasma of unbound fermionic particles or a gas of composite bosons called excitons. Usually, the exciton phase is associated with low temperatures. In atomically thin transition metal dichalcogenide semiconductors, excitons are particularly important even at room temperature due to strong Coulomb interaction and a large exciton density of states. Using state-of-the-art many-body theory, we show that the thermodynamic fission–fusion balance of excitons and electron-hole plasma can be efficiently tuned via the dielectric environment as well as charge carrier doping. We propose the observation of these effects by studying exciton satellites in photoemission and tunneling spectroscopy, which present direct solid-state counterparts of high-energy collider experiments on the induced fission of composite particles.

[1] Institut für Theoretische Physik, Universität Bremen, P.O. Box 330 440, 28334 Bremen, Germany. [2] Bremen Center for Computational Materials Science, Universität Bremen, TAB-Gebäude, Am Fallturm 1, 28359 Bremen, Germany. [3] MAPEX Center for Materials and Processes, Universität Bremen, 28359 Bremen, Germany. [4] Present address: Department of Physics and Astronomy, University of Southern California, 825 Bloom Walk, ACB 439, Los Angeles, CA 90089-0484, USA. Correspondence and requests for materials should be addressed to A.S. (email: asteinhoff@itp.uni-bremen.de)

The interplay of excitons and unbound electron-hole pairs is at the heart of excited-semiconductor physics. Due to exceptionally strong electron-hole Coulomb interaction and a naturally high sensitivity to spectroscopic methods, atomically thin semiconductors from the class of transition metal dichalcogenides (TMDCs) are perfectly suited to study the fission of excitons. The latter present a prominent realization of composite bosons formed by fermionic constituents and therefore provide insight beyond the specific material class of two-dimensional semiconductors. The prominent role of excitons in the optical properties of TMDCs suggests an interpretation of experimental results as well as theoretical prediction in terms of excitons rather than unbound electrons and holes[1–5]. On the other hand, it is well known that, at a certain excitation density of electron-hole pairs, the Mott transition is observed[6–8]. Here a phase where excitons and unbound carriers can coexist evolves into a fully ionized electron-hole plasma.

Since excitons are more or less neutral compound bosonic particles, many-particle renormalization and screening effects in an exciton gas are very different from those in a plasma of unbound electrons and holes, which we refer to in the following as quasi-free particles. For this reason, it is highly desirable to quantify the relative importance of excitonic and plasma effects over a wide range of electron-hole excitation densities and to learn how it can be manipulated from the outside. It has already been suggested to tune exciton binding energies by electrical doping[8] and some effort has been devoted to study the influence of dielectric screening on excitons in TMDC semiconductors[9–13]. In the past, a powerful scheme has been developed to theoretically describe the balance between fission and fusion of excitons and quasi-free particles, also termed ionization equilibrium[14–19], with applications to atomic plasmas and highly excited semiconductors. The scheme relies on the assumption of a quasi-equilibrium between plasma and excitons being established before electron-hole recombination sets in. This is supported by ultrafast equilibration due to efficient carrier–carrier[20] and carrier–phonon interaction[5] as well as exciton formation[21,22] after optical excitation, see ref. [23] for a review.

Experimental verification of the ionization equilibrium has been achieved in GaAs quantum wells using THz spectroscopy to probe transitions between 1s- and 2p-exciton states[23,24]. A similar technique in the mid-infrared range has been applied recently to monolayer WSe$_2$[22]. Alternatively, the fractions of excitons and plasma can be determined from their contributions to photoluminescence spectra[25] in combination with additional photoluminescence simulations.

Here we show that a phase largely dominated by excitons at elevated excitation densities and its abrupt transformation into an electron-hole plasma at the Mott transition are found at room temperature in monolayer TMDC materials. At low densities, exciton fission due to entropy effects is predicted. We demonstrate that the thermodynamical balance between fission and fusion of excitons and quasi-free particles can be directly manipulated by the choice of dielectric environment as well as charge carrier doping. We also suggest that new ways to quantify the fission–fusion balance are angular-resolved photoemission spectroscopy (ARPES) and scanning tunneling spectroscopy (STS). The observation of excitons by these methods can at the same time be understood as fission of composite bosons, induced by incident photons or applied voltage. Photoemission spectroscopy also gives access to the extent of exciton wave functions via the structure of exciton satellites. To obtain quantitative results for the materials MX$_2$ (M = W,Mo and X = S,Se), we build on the theory of ionization equilibrium, combining it for the first time with material-realistic band structure and Coulomb matrix element calculations that enable us also to study the influence of the dielectric environment. Beyond frequency-dependent plasma screening, we additionally include screening due to excitons. The latter is shown to be relevant although it has neither been discussed before in the context of ionization equilibrium nor for two-dimensional materials.

## Results

**Spectral functions and exciton satellites.** To examine the equilibrium properties of excited carriers in TMDCs, we use the quantum-statistical expression for the carrier density $n_a$ of the species $a$, which can be electrons or holes, as a function of temperature $T$ and chemical potential $\mu_a$ as a starting point:

$$n_a(\mu_a, T) = \frac{i\hbar}{\mathcal{A}} \int_{-\infty}^{\infty} \frac{d\omega}{2\pi} \sum_{\mathbf{k}\sigma} f^a(\omega) A_{\mathbf{k}\sigma}^a(\omega). \quad (1)$$

$f^a(\omega)$ denotes the Fermi distribution function depending on $\mu_a$ and $T$, $\mathcal{A}$ is the crystal area and $A_{\mathbf{k}\sigma}^a(\omega) = 2i\mathrm{Im}G_{\mathbf{k}\sigma}^{\mathrm{ret},a}(\omega)$ is the spectral function of the single-particle state $|\mathbf{k}\sigma a\rangle$ related to the retarded single-particle Green's function

$$G_{\mathbf{k}\sigma}^{\mathrm{ret},a}(\omega) = \frac{1}{\hbar\omega - \varepsilon_{\mathbf{k}\sigma}^a - \Sigma_{\mathbf{k}\sigma}^{\mathrm{ret},a}(\omega)}. \quad (2)$$

The self-energy $\Sigma_{\mathbf{k}\sigma}^{\mathrm{ret},a}(\omega)$ accounts for many-particle effects giving rise to renormalizations of the single-particle band structure $\varepsilon_{\mathbf{k}\sigma}^a$ as well as contributions of bound states. For a given self-energy, the inversion of Eq. (1) yields the chemical potential $\mu_a(n_a, T)$ for each species and therefore any thermodynamic property of the system in the grand canonical formulation. As we describe in detail in the "Methods" section, by using a T-matrix self-energy in screened ladder approximation and assuming small quasi-particle damping, we obtain a spectral function $A^a(\omega)$ in the so-called extended quasi-particle approximation. It exhibits poles for quasi-free and bound carriers as shown in Fig. 1. From the multiple valleys in the single-particle band structure of electrons and holes (Fig. 1a), a rich spectrum of bound states emerges (Fig. 1b) which contains a variety of dark excitons with large total momentum $Q$ besides the bright $K$-valley excitons commonly referred to as A and B. The dark excitons, though playing a minor role in optical experiments, are essential to the description of the ionization equilibrium. Various bound states are reflected in the low-energy satellites of the single-particle spectral function. Excitonic contributions are expected to be observed in experiments that are sensitive to these spectral properties. In ARPES[26], momentum-resolved images of the electron spectral function comparable to Fig. 1c are obtained, which are weighted with Fermi distribution functions that are defined by the chemical potential $\mu_e$ and temperature $T$. In quasi-equilibrium, the measured intensity is given by

$$I_{\mathbf{k}}(\omega) \propto \sum_{\sigma} f^e(\omega) A_{\mathbf{k}\sigma}^e(\omega). \quad (3)$$

For fixed quasi-momentum $\mathbf{k}$, as shown in Fig. 1e, this is typically referred to as energy distribution curve. On the other hand, STS[27] probes the local density of states and thus momentum-averaged spectral functions of electrons and holes, which are displayed in Fig. 1c, d. We therefore propose to use these well-established experimental techniques to spectrally distinguish between excitons and quasi-free carriers. To this end, spectroscopy has to be combined with prior optical excitation of the semiconductor, requiring time-resolved experiments. Time-resolved ARPES with high temporal and spectral resolution has recently been applied to two-dimensional materials, which prove to be highly sensitive to this method[28–30]. In addition, atomically thin semiconductors, that are available in ever-improving sample quality, have the

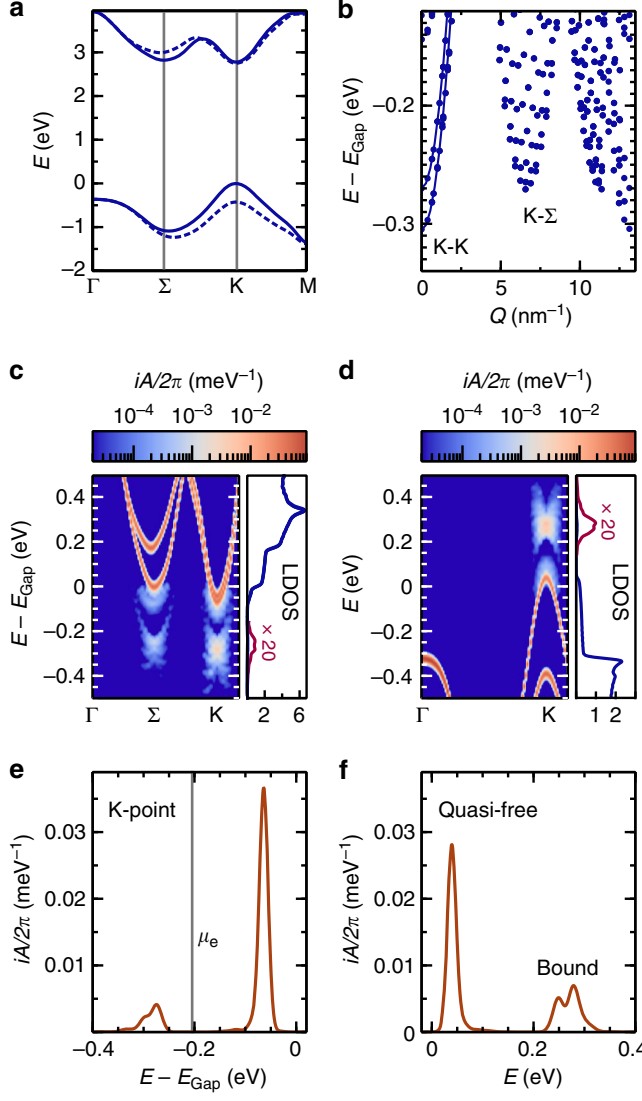

**Ionization equilibrium and Mott transition.** The spectral function in extended quasi-particle approximation is given by

$$A_{\mathbf{k}\sigma}^a(\omega) = -2\pi i \delta(\hbar\omega - E_{\mathbf{k}\sigma}^a)\left(1 - Z_{\mathbf{k}\sigma}^a\right)$$

$$-2\pi i \Gamma_{\mathbf{k}\sigma}^a(\omega), \tag{4}$$

where $\Gamma_{\mathbf{k}\sigma}^a(\omega)$ and the renormalization factor $Z_{\mathbf{k}\sigma}^a$ account for two-particle states, as discussed in detail in the "Methods" section. According to Eq. (1), the spectral function can be used to separate the total electron and hole density ($a = $ e, h),

$$n_a = n_{\text{free}}^a + n_X, \tag{5}$$

into contributions from quasi-free carriers and from carriers bound as excitons. The excitons, defined by two-particle energies below the quasi-particle band gap, are approximately treated as bosons. Hence, the properties of the excited semiconductor at a given temperature and excitation density are described by the density of electrons $n_e$, the density of holes $n_h$ and the density of excitons $n_X$. The degree of ionization of the excited carriers

$$\alpha_a = \frac{n_{\text{free}}^a}{n_a} \tag{6}$$

will be established as a result of the ionization equilibrium between electrons, holes and excitons. While for optical excitation, equal densities of electrons and holes are generated, we distinguish here between electron and hole ionization to also include the effect of carrier doping, where electron and hole densities are different.

Using single-particle band structures and bound-state spectra, which are determined as discussed in the "Methods" section, we solve Eq. (5) numerically to obtain the degree of ionization $\alpha_a$ in various TMDC materials under different experimental conditions. The results are collected in Fig. 2 and exhibit the behaviour of the ionization degree as a function of the excitation density. There are different regimes of ionization to be observed. At high excitation densities between $3 \times 10^{12}$ cm$^{-2}$ and $1 \times 10^{13}$ cm$^{-2}$, depending on experimental parameters, efficient screening and many-particle renormalizations lead to a full ionization of excited carriers, which is known as Mott effect. At lower densities around $n_a = 1 \times 10^{12}$ cm$^{-2}$, excitons dominate the physical properties of TMDCs for the parameters studied here due to the large exciton-binding energies and a density of states with dominant contributions from dark excitons. Bright excitons with very small momenta that are optically active make up only a tiny fraction of the total exciton density, as illustrated in Fig. 3. The density of bright excitons is smaller than the total exciton density by about five orders of magnitude in MoS$_2$ and six orders of magnitude in WSe$_2$ over the whole range of excitation densities below the Mott transition. Although only bright excitons directly recombine, excitons with larger momentum can relax via efficient exciton–phonon interaction[5] and refill optically active states, thereby representing an efficient reservoir for bright excitons. In Fig. 3, we also provide the density of all intra-valley excitons formed by carriers with equal spins in the $K$ and $K'$ valleys. These excitons make up a much larger fraction of the total exciton density, while still most of the excitons feature electrons and holes with different spins and/or valleys.

As Fig. 2b shows, an efficient tuning knob for the degree of ionization is the dielectric screening due to the environment, which can change over a wide range depending on the experimental situation or device realization in which the TMDC

**Fig. 1** Spectral properties of excited carriers in monolayer WS$_2$. **a** Band structure of freestanding WS$_2$ as obtained from a G$_0$W$_0$ calculation at zero excitation density, including spin-orbit interaction. **b** Bound-state spectrum for WS$_2$ on SiO$_2$ substrate relative to the quasi-particle gap at zero excitation density over the modulus of total exciton momentum $Q$ as obtained from a Bethe–Salpeter equation, see Eq. (16) in the "Methods" section. Excitons involving the $\Gamma$, $K'$ and $\Sigma'$ valleys are included but not marked explicitly. The lines serve as guide to the eye for the $K$-exciton dispersion. **c** Electron spectral function in extended quasi-particle approximation at $T = 300$ K and excitation density $n_a = 3.2 \times 10^{12}$ cm$^{-2}$ showing resonances from bound and quasi-free particles in momentum-resolved representation and as normalized local density of states. Energies are measured relative to the quasi-particle band gap at zero excitation density. A phenomenological Gaussian broadening of 10 meV (HWHM) is applied. **d** Hole spectral function in extended quasi-particle approximation. **e** Electron spectral function as in **c** for spin-down electrons at the $K$-point. The vertical line marks the electron chemical potential. **f** Hole spectral function as in **d** for spin-up holes at the $K$-point

advantage of a large spectral separation of excitonic and quasi-particle signatures, which is well above the available energy resolution. These modern developments open the possibility to quantify the degree of exciton ionization and to access the extent of exciton wave functions, which are encoded in the structure of the exciton satellites.

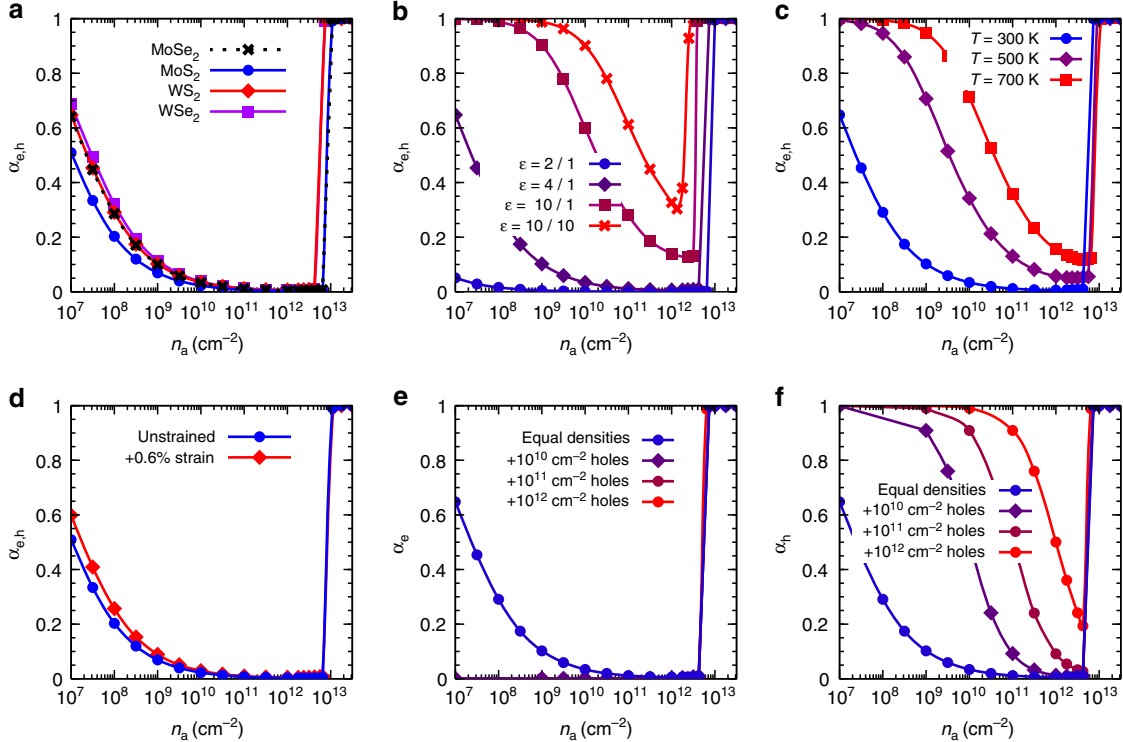

**Fig. 2** Degree of ionization as a function of excitation density. $n_a = n^a_{free} + n_X$ denotes the total density of electrons or holes. In general, a fully ionized plasma ($\alpha_a = 1$) is found above the Mott density, excitons dominate ($\alpha_a \ll 1$) below the Mott density and ionization appears again at very low densities. If not stated otherwise, the temperature is 300 K. **a** Comparison of different TMDC materials on $SiO_2$ substrate. **b** Comparison of $WS_2$ in different dielectric environments with dielectric constant $\varepsilon$ for the bottom/top surrounding of the monolayer. **c** Comparison of $WS_2$ on $SiO_2$ substrate for different temperatures. **d** Influence of strain for $MoS_2$ on $SiO_2$ substrate. The unstrained layer corresponds to a lattice constant $a = 3.18$ Å, while tensile biaxial strain is simulated by $a = 3.20$ Å. **e** Fraction of ionized electrons for different levels of hole doping of $WS_2$ on $SiO_2$ substrate. **f** Fraction of ionized holes for different levels of hole doping of $WS_2$ on $SiO_2$ substrate

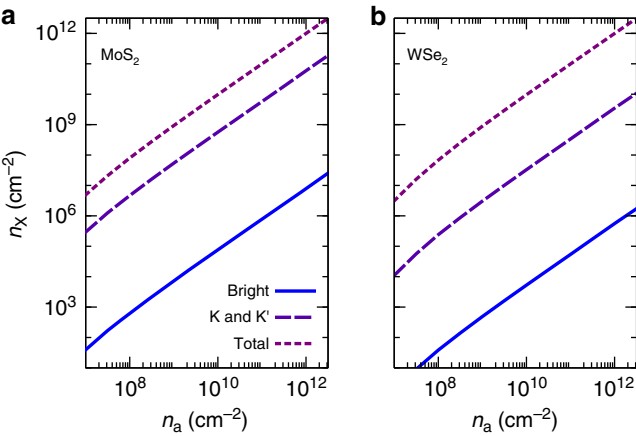

**Fig. 3** Comparison of the relative importance of bright and dark excitons. The density of bright excitons that fulfill energy and momentum conservation in radiative recombination processes, the dark exciton density due to intra-valley excitations in the vicinity of K and K', as well as the total exciton density vs. excitation density are shown for **a** $MoS_2$ and **b** $WSe_2$ on $SiO_2$ substrate at $T = 300$ K

monolayer is used. The reason is the strong impact of dielectric screening on the exciton binding energies. Typical examples for substrates are Borofloat ($\varepsilon = 2$), $SiO_2$ ($\varepsilon = 4$) and sapphire ($\varepsilon = 10$). The dielectric constant of the environment on top of the

monolayer is often given by the vacuum value. On the other hand, in devices the TMDC monolayer is usually fully encapsulated by dielectric material. As an example we consider a full dielectric enclosure with $\varepsilon = 10$, which might be either sapphire or additional layers of TMDC material in a vertical heterostructure whose main influence on the excitons is the dielectric screening[31]. We find that the minimal degree of ionization can be tuned from below 0.1% (99.9% excitons) for weak dielectric screening to about 30% for strong screening, while the Mott density is lowered at the same time by about a factor of 3. The second important parameter, that is relevant for applications of TMDC monolayers, is the doping with additional carriers which might be either intrinsic or induced by external electric fields in a capacitor structure. Here the fractions of ionized electrons and holes, $\alpha_e$ and $\alpha_h$, are discussed separately as the densities of the species are not equal anymore. We consider hole doping of $WS_2$, but similar results are expected in case of electron doping. According to Fig. 2e, f, even for weak doping the minority carriers are practically all bound as excitons below the Mott transition. On the other hand, for higher doping levels an increasing fraction of majority carriers exists as quasi-free plasma due to missing partners for exciton formation. As a function of minority-carrier density, the Mott transition is lowered by about the density of doped excess carriers. From this, we conclude that at doping levels above $10^{13}$ cm$^{-2}$ neither dark nor bright excitons will exist in any case. This is facilitated by the experimental estimate for doping-induced ionization at several $10^{13}$ cm$^{-2}$ [8]. As Fig. 2c shows, the temperature is another crucial parameter, which can vary in experiments or devices due to heating of the

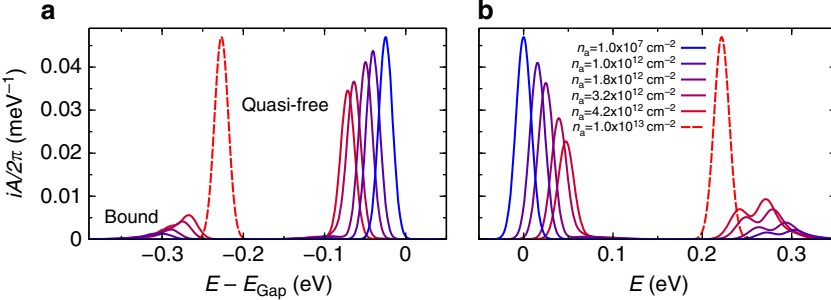

**Fig. 4** Energy distribution and relative weight of the exciton and quasi-free-carrier contributions represented by the spectral function. For increasing excitation density and a temperature of 300 K, the spectral function is determined in extended quasi-particle approximation for spin-down electrons (**a**) and spin-up holes (**b**) at the $K$-point in WS$_2$ on SiO$_2$ substrate. A phenomenological Gaussian broadening of 10 meV (HWHM) is included. Electron energies are measured relative to the quasi-particle band gap at zero excitation density. Hole energies are shown as valence-band energies to match Fig. 1f

active material under strong optical or electrical pumping. This effect has been used to explain the observed exciton-to-plasma ratio in monolayer WSe$_2$ in ref. [22]. At room temperature and even at elevated temperatures up to 700 K, excitons clearly dominate below the Mott transition. At the same time, the Mott density slightly increases with temperature due to weaker renormalizations of the quasi-particle gap. It turns out that strain is no efficient tuning knob as both, bright and dark excitons contribute to the ionization equilibrium, although bright excitons are preferred in moderately tensile-strained TMDCs, see Supplementary Fig. 2c. A comparison of different TMDC materials shows that excitons are slightly more important in molybdenum- than in tungsten-based TMDCs due to the larger binding energies, which leads to higher Mott densities.

When approaching the Mott density from the low-density side, many-particle renormalizations, as given by Eq. (25) in the "Methods" section, become increasingly important. Exchange interaction and efficient screening due to free carriers as well as excitons reduce the quasi-particle band gap and the exciton binding energies. More and more excitons are ionized, which leads to an increase of efficient free-carrier screening and thereby to a self-amplification of the ionization effect until all excitons are dissociated into an electron-hole plasma and the degree of ionization becomes $\alpha_a = 1$. Note that $\alpha_a$ includes not only bright but also dark excitons with large total momenta for example between $K$ and $\Sigma$ valleys. Those excitons may have larger binding energies, as also discussed in ref. [32], and they are slightly more stable against ionization than bright excitons visible in an optical experiment. Fig. 4 shows an illustration of the Mott effect in terms of the spectral functions in extended quasi-particle approximation, which contain both exciton and quasi-free-particle signatures. At very low excitation densities, the only spectral contribution stems from quasi-free carriers at the band edge. With increasing density, the quasi-particle peak is shifted to lower energies due to many-particle renormalizations. At the same time, spectral weight is transferred from the unbound quasi-particle to the bound-state peaks as exciton populations increase, see the explicit expression of the spectral function in Eq. (23). The appearance of several exciton satellites in the hole spectral function is due to different bound states involving electrons either in the $K$- or $\Sigma$-valleys, see Fig. 1b. The energetic position of a bound-state peak in the spectral function of carrier $a$ is given by the difference of the corresponding bound-state energy $E^{ab}$, which is an eigenenergy of the Bethe–Salpeter Eq. (16), and the energy of the second carrier $b$ involved in the bound state. The bound resonance might therefore be interpreted as an effective ionization energy of the actual carrier $a$ with respect to its energy in the quasi-particle band structure. Both the quasi-particle band

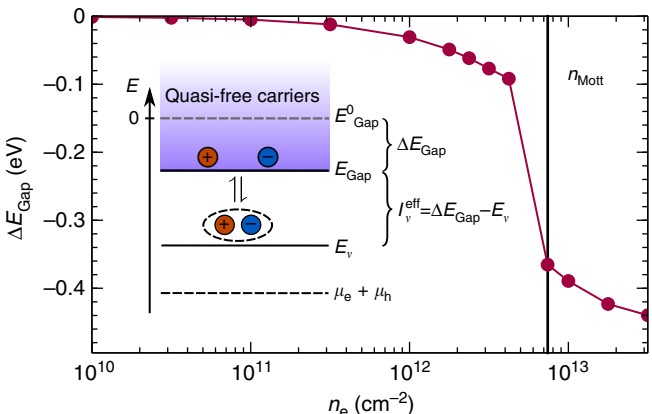

**Fig. 5** Ionization equilibrium and quasi-particle band-gap renormalization. The band-gap renormalization $\Delta E_{Gap}$ for increasing excitation density is depicted for spin-up carriers at the $K$-point in WS$_2$ on SiO$_2$ substrate at $T = 300$ K. The vertical black line marks the Mott transition. The inset is a schematic of the ionization equilibrium introducing the quasi-particle band gap at zero excitation density $E^0_{Gap}$, which is conventionally set to zero energy. All quantities including the bound-state energies $E_\nu$ and the sum of electron and hole chemical potentials are measured relative to $E^0_{Gap}$

structure and the effective ionization energies are observable in experiments that are sensitive to the single-particle spectral function such as ARPES and STS. Despite the fact that the amplitudes of bound-state resonances in the spectral functions are relatively small, observables like the carrier density, Eq. (1), and the photoemission intensity, Eq. (3), involve weighting with a Fermi function that strongly favors the low-energy resonances over the quasi-free contribution. With increasing excitation density, quasi-particle and excitonic resonances approach each other until at the Mott density all excitons are ionized and only a quasi-particle peak of unbound carriers remains. Figure 5 shows the reduction of the quasi-particle gap until the Mott transition appears around $n_a = 8 \times 10^{12}$ cm$^{-2}$.

An alternative picture of the interacting electrons and holes, which is consistent with the extended quasi-particle approximation, is the so-called "chemical picture", in which excitons are considered as a new particle species besides electrons and holes[18,19]. They are characterized by a chemical potential

$$\mu_{X,\nu} = \mu_e + \mu_h - E_\nu,\qquad(7)$$

with bound-state energies $E_\nu$ that are given by the relative motion of electron and hole, and an ideal Bose distribution function. In

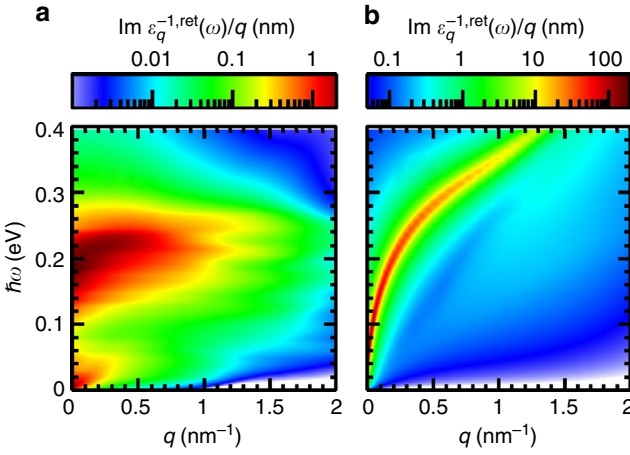

**Fig. 6** Different screening properties of excitons and quasi-free carriers. Plasmon spectral function given by the imaginary part of the inverse dielectric function, including the 2-d-Coulomb singularity. In the calculation, the contributions due to dynamical plasma and excitonic screening are included. The results are depicted for momenta along the contour Γ-K in WS$_2$ on SiO$_2$ substrate at $T = 300$ K. A quasi-particle broadening of 10 meV is used. The carrier densities are (**a**) $n_{free} = 2.7 \times 10^{10}$ cm$^{-2}$, $n_X = 3.1 \times 10^{12}$ cm$^{-2}$ and (**b**) $n_{free} = 1.8 \times 10^{13}$ cm$^{-2}$, $n_X = 0$ corresponding to points on the ionization equilibrium curve in Fig. 2 in the exciton-dominated and the plasma-dominated regime, respectively

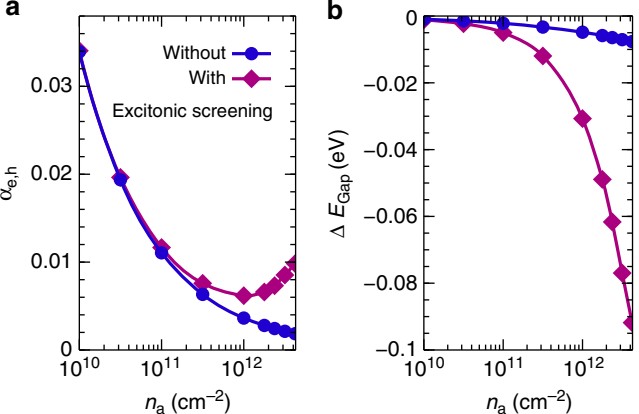

**Fig. 7** Influence of excitonic screening on the ionization equilibrium. **a** Degree of ionization with and without excitonic screening as a function of excitation density $n_a = n_{free}^a + n_X$ below the Mott transition in WS$_2$ on a SiO$_2$ substrate at $T = 300$ K. **b** Corresponding band-gap renormalization at the $K$-point

the chemical picture, solving Eq. (5) corresponds to an adaption of the chemical potentials of the different particle species, namely electrons, holes and excitons, as in a chemical reaction. These considerations are consistent with the theory based on spectral functions, that we use to obtain all numerical results presented in this paper. Only for the purpose of illustration, we simplify the theory considering the nondegenerate case ($f^a(E_{\mathbf{k}\sigma}^a) \ll 1$), a single band-structure valley for electrons and holes each and momentum-independent band-structure renormalizations. Then a Saha equation can be formulated that determines the degree of ionization:

$$\frac{n_X}{n_e n_h} \propto e^{\beta I_\nu^{eff}}. \qquad (8)$$

In analogy to the usual mass action law, $I_\nu^{eff} = \Delta E_{Gap} - E_\nu$ can be interpreted as an effective ionization potential of excitons that corresponds to the exciton binding energy, see also the inset in Fig. 5. It is obvious from Saha's equation that a large exciton binding energy and low temperature favor the formation of excitons vs. the dissociation into an unbound electron-hole plasma. The ionization potential depends on excitation density as a consequence of the excitation-induced lowering of the band continuum edge given by $\Delta E_{Gap}$ and the shift of the bound-state energy $E_\nu$. The bound-state shift on the other hand is a net result of band-gap shrinkage, screening of exciton binding energy and Pauli blocking[33] and is much weaker than the band-gap shift due to compensation effects. In the end, the ionization potential is lowered with increasing excitation density until at $I_\nu^{eff} = 0$ the bound state vanishes and merges with the continuum edge, which is the Mott effect.

A striking observation in Fig. 2 is the degree of ionization approaching unity at low excitation densities, which is somewhat counter-intuitive but can be understood from a thermodynamical point of view. The potential that is minimized by the many-particle system is the free energy $F = U - TS$. At low densities and fixed temperature, the entropy $S$ gained by a dissociation of an

exciton into two separate particles overcompensates the reduction of internal energy $U$ by the exciton binding energy $E_B$. Hence, the so-called entropy ionization already discussed by Mock et al.[34] is connected to the huge phase space available for quasi-free carriers in the low-density limit. We may clarify this using the entropy of an ideal gas with $N$ particles in a volume $V$ as given by the Sackur-Tetrode equation:

$$S = N k_B \left[ \ln\left(\frac{V}{N} c(T)\right) + \frac{5}{2} \right], \qquad (9)$$

where $c(T)$ is a temperature-dependent parameter. Obviously, the dissociation of an exciton gas ($N$ particles) into a free electron-hole plasma ($2N$ particles) yields the entropy $\Delta S = N k_B \ln(n^{-1})$ with $n = N/V$ up to some additive constant. It follows that the critical density $n_{crit}$ below which the free energy is dominated by entropy essentially scales as $\exp(-E_B/k_B T)$ with temperature.

**Excitonic screening.** In the spirit of the extended quasi-particle approximation to the spectral function, there are two types of contributions to excited-carrier screening of the Coulomb interaction, the metal-like free-carrier screening and dipolar screening due to bound excitons. The screening can be characterized by the plasmon spectral function, see Eq. (29), that contains excitations in the interacting electron-hole plasma as poles in the $q$-$\omega$-plane, see Fig. 6. In the exciton-dominated regime shown in Fig. 6a, besides the usual 2-d free-carrier contribution at small energies and small momenta, a broad resonance above 150 meV appears. It stems from transitions between 1s- and 2s- as well as 2p-like exciton states, see Fig. 1b, and also from comparable transitions between exciton states with large momenta. There are contributions at smaller energies as well that can not be as easily distinguished from free-carrier screening. At large densities beyond the Mott transition, the plasmon spectral function shows a pronounced peak structure with a square-root-like behaviour at small momenta, which has been discussed for TMDCs in ref. [35] and which is typical for a two-dimensional electron gas[36]. Excitons are expected to be much less polarizable than a free electron-hole plasma and, hence, contribute less to screening. Nevertheless, at elevated excitation densities with more than 99% of carriers bound as excitons, their contribution is significant. We demonstrate this by comparing the results for the degree of ionization $\alpha_a$

with and without excitonic screening included using the example of $WS_2$ on $SiO_2$ substrate. The results are shown in Fig. 7a. Excitonic screening efficiently reduces the ionization potential of excitons at elevated excitation densities, thereby triggering the transition to an ionized plasma, which is reflected by an increase of the degree of ionized carriers. This mechanism of exciton fission is absent when excitonic screening is neglected, thus leading to an ongoing decrease of the ionized-carrier fraction when coming from the low-density side of the ionization curves. From the many-particle renormalization of the band gap caused by free-carrier and excitonic screening as shown in Fig. 7b, we deduce that in monolayer TMDC semiconductors excitonic screening is less efficient by two orders of magnitude for comparable excitation densities. As the plasmon spectral function is directly observable by electron energy loss spectroscopy[37], we suggest to use this technique to explore exciton signatures in the dielectric function eperimentally.

## Discussion

Having the composite nature of excitons in mind, photoemission and tunneling spectroscopy on excitons can be seen as semiconductor analogue to induced fission of bosonic particles into their constituents in high-energy collider experiments. Examples include the photodesintegration of deuterium[38] and the photofission of heavy nuclei[39], where nuclear forces instead of Coulomb forces have to be overcome. In time-resolved ARPES, excitons are fissured by photons in the eV range and momentum and energy of electrons as fission products are detected, thereby revealing information about the internal structure of the excitons. In STS, the role of photons as external probe is assumed by a voltage and the spectrum is an average over momentum states. TMDC monolayers offer the unique possibility to optically address different band-structure valleys selectively[40], which we expect to be reflected by the exciton satellites in photoemission studies at short time delays after circularly polarized excitation.

Although the extended quasi-particle approximation and the chemical picture as applied in this paper are very descriptive, we have to be aware of their limitations. Firstly, the approach relies on the assumption of a quasi-equilibrium of both types of carriers, excitons and quasi-free plasma. A mechanism that yields corrections to this quasi-equilibrium picture is the electron-hole recombination, either radiative or nonradiative, that reduces the excitation density on a ps time scale[22]. Given the fact that the relaxation and equilibration of excitons and quasi-free carriers are much faster than this[5,20,41], empty states are immediately refilled and the system practically remains in quasi-equilibrium, where the exciton-plasma balance adiabatically adapts to the time-dependent density. This picture is also applied in ref. [22], where a ratio between excitons and plasma is assigned to the experimental data at each time during carrier recombination.

A rather fundamental discussion is concerned with the Mott transition as a first-order phase transition between an exciton gas and a fully ionized electron-hole plasma[18,19]. The phase transition would be connected to an instability of thermodynamic functions that manifests itself in an ambiguity of $\alpha_a$ in a certain region below the Mott density. Due to excitation-induced broadening of the two-particle states, which is assumed small in our approach, and the shrinkage of the ionization potential towards the Mott transition, quasi-free and bound carriers cannot really be separated in this density regime. We avoid this regime as a more sophisticated theory including full spectral functions and exciton-exciton interaction would be required. Also, screening in a correlated many-particle system near the Mott transition is an intricate problem[42].

A prominent feature of TMDC semiconductors is the formation of trions, which could in principle be included as additional particle species in the spirit of the chemical picture[17]. In practice, obtaining bound-trion spectra on the same footing as excitons is a very challenging task on its own, which is beyond the scope of this paper. Qualitatively, one can expect a coexistence of all three phases at room temperature, where some of the exciton population will be drawn to bound-trion states. As trion binding energies are typically in the range of 30 meV[43], which is an order of magnitude less than exciton binding energies, this thermal redistribution of population will be much smaller than between excitons and plasma.

Beyond the monolayer limit, TMDC semiconductors can be used to construct bilayer systems[44] that allow for the formation of spatially indirect, long-lived excitons. These systems are expected to be well suited to realize exotic composite-boson phases like condensates at record-high temperatures. Our material-realistic approach to the ionization equilibrium may be extended to include exciton condensates and used to study the complex phase diagrams of bilayer systems. Here, in analogy to electron-hole GaAs/InGaAs bilayers, signatures of exciton fission may also be observed in the temperature dependence of the Coulomb drag effect[45].

In conclusion, the ionization equilibrium between the fission and fusion of excitons and electron-hole pairs in monolayer TMDC semiconductors has been studied for various material as well as experimentally and device-relevant external parameters on the basis of an ab initio description of the electronic band structure and Coulomb interaction. We observe entropy ionization of excitons at low excitation densities and a Mott transition to a fully ionized plasma at high densities between $3 \times 10^{12}$ cm$^{-2}$ and $1 \times 10^{13}$ cm$^{-2}$, depending on experimental parameters. Below the Mott transition, excitons become dominant in all cases with maximal fractions of excitons between 70 and >99.9%. The most efficient tuning knobs are dielectric screening of the Coulomb interaction via the choice of dielectric environment and carrier doping that can induce complete ionization above a level of $10^{13}$ cm$^{-2}$. Moreover, we find that excitonic screening, although two orders of magnitude less efficient than free-carrier screening at comparable excitation densities, plays an important role in the description of ionization equilibrium. We suggest that fingerprints of excitonic contributions can be observed in ARPES and STS experiments, which are sensitive to the single-particle spectral functions, thus containing information about the degree of exciton fission and the extent of exciton wave functions in reciprocal space.

## Methods

**Theory of ionization equilibrium**. We start from the general expression for the carrier density (1) and the spectral function

$$A^a_{\mathbf{k}\sigma}(\omega) = 2i\mathrm{Im} \frac{1}{\hbar\omega - \varepsilon^a_{\mathbf{k}\sigma} - \Sigma^{\mathrm{ret},a}_{\mathbf{k}\sigma}(\omega)}. \tag{10}$$

In the limit of small quasi-particle damping $\mathrm{Im}\,\Sigma^{\mathrm{ret},a} \ll \mathrm{Re}\,\Sigma^{\mathrm{ret},a}$, the spectral function can be expanded in linear order of $\mathrm{Im}\,\Sigma^{\mathrm{ret},a}$ yielding the carrier density in so-called extended quasi-particle approximation

$$
\begin{aligned}
n_a(\mu_a, T) = &\frac{1}{\mathcal{A}} \sum_{\mathbf{k}\sigma} f^a(E^a_{\mathbf{k}\sigma}) \\
&- \frac{1}{\mathcal{A}} \sum_{\mathbf{k}\sigma} \int_{-\infty}^{\infty} \frac{d\omega}{2\pi} \frac{2}{\hbar} \mathrm{Im}\,\Sigma^{\mathrm{ret},a}_{\mathbf{k}\sigma}(\omega) \left[ f^a(E^a_{\mathbf{k}\sigma}) - f^a(\omega) \right] \\
&\times \frac{d}{d\omega} \frac{\mathcal{P}}{\omega - E^a_{\mathbf{k}\sigma}/\hbar} \\
= &\, n^{\mathrm{QP}}_a + n^{\mathrm{corr}}_a,
\end{aligned} \tag{11}
$$

where the quasi-particle energy $E^a_{\mathbf{k}\sigma}$ is given by $E^a_{\mathbf{k}\sigma} = \varepsilon^a_{\mathbf{k}\sigma} + \mathrm{Re}\,\Sigma^{\mathrm{ret},a}_{\mathbf{k}\sigma}(E^a_{\mathbf{k}\sigma})$ and $\mathcal{P}$ denotes the Cauchy principal value[14,19]. The total density is divided into

contributions from quasi-free particles and correlated particles, the latter being either in bound or scattering many-particle states.

The spectral function in extended quasi-particle approximation corresponding to this separation into free and correlated carriers is given by

$$A_{\mathbf{k}\sigma}^{a}(\omega) = -2\pi i\delta(\hbar\omega - E_{\mathbf{k}\sigma}^{a})\left(1 - Z_{\mathbf{k}\sigma}^{a}\right)$$

$$-2\pi i\Gamma_{\mathbf{k}\sigma}^{a}(\omega),$$

(12)

with $\Gamma_{\mathbf{k}\sigma}^{a}(\omega) = \mathrm{Im}\,\Sigma_{\mathbf{k}\sigma}^{\mathrm{ret},a}(\omega)\frac{1}{\pi}\frac{\mathrm{d}}{\mathrm{d}\hbar\omega}\frac{\mathcal{P}}{\hbar\omega - E_{\mathbf{k}\sigma}^{a}}$ and the renormalization factor $Z_{\mathbf{k}\sigma}^{a} = \int\mathrm{d}\hbar\omega\,\Gamma_{\mathbf{k}\sigma}^{a}(\omega)$. The first term describes quasi-free particles at renormalized energies. Their spectral weight is reduced according to the renormalization factor to account for correlated carriers, which are spectrally described by the second term.

To evaluate the expressions (11) and (12), we have to choose an approximation for the self-energy $\Sigma^{\mathrm{ret},a}(\omega)$. The real and imaginary parts of $\Sigma$ determine the quasi-particle energies and the correlated part of the carrier density, respectively. An appropriate choice is the screened ladder approximation[14,17,46] $\Sigma(\omega) = \Sigma^{\mathrm{H}} + \Sigma^{\mathrm{GW}}(\omega) + \Sigma^{\mathrm{T}}(\omega)$ that takes into account screening of Coulomb interaction due to excited carriers as well as the formation of bound two-particle states and consists of Hartree, GW and T-matrix contributions. We assume that renormalizations due to the Hartree self-energy are small compared to exchange and correlation effects. In the T-matrix contribution, we neglect exchange terms and assume static screening so that the T-matrix depends only on one instead of three frequency arguments. Thus, we obtain for the imaginary part of the self-energy using the generalized Kadanoff–Baym ansatz[18]:

$$\mathrm{Im}\,\Sigma_{\mathbf{k}\sigma}^{\mathrm{ret},a}(\omega) = -\frac{1}{\mathcal{A}}\sum_{\mathbf{k}b}V_{\mathbf{k}'\mathbf{k}\mathbf{k}'}^{ab}\,\mathrm{Im}\,\varepsilon_{\mathbf{k}-\mathbf{k}'}^{-1,\mathrm{ret}}(\omega - E_{\mathbf{k}'\sigma}^{b}/\hbar)$$

$$\times\left[f^{b}(E_{\mathbf{k}'\sigma}^{b}) + n^{\mathrm{B}}(E_{\mathbf{k}'\sigma}^{b} - \hbar\omega)\right]$$

$$+\frac{1}{\mathcal{A}}\sum_{\mathbf{k}b\sigma'}\mathrm{Im}\,T_{\mathbf{k}\mathbf{k}'\sigma\sigma'}^{'',\mathrm{ret},ab}(\omega + E_{\mathbf{k}'\sigma'}^{b}/\hbar)$$

$$\times\left[f^{b}(E_{\mathbf{k}'\sigma'}^{b}) + n_{ab}^{\mathrm{B}}(\hbar\omega + E_{\mathbf{k}'\sigma'}^{b})\right].$$

(13)

Here we applied thermal equilibrium relations for the screened Coulomb interaction[18]:

$$V_{\mathbf{k}\mathbf{k}'\mathbf{k}}^{\mathrm{S},<,ab}(\omega) = n^{\mathrm{B}}(\omega)V_{\mathbf{k}\mathbf{k}'\mathbf{k}}^{ab}2i\mathrm{Im}\,\varepsilon_{\mathbf{k}-\mathbf{k}'}^{-1,\mathrm{ret}}(\omega),$$

$$V_{\mathbf{k}\mathbf{k}'\mathbf{k}}^{\mathrm{S},>,ab}(\omega) = (1 + n^{\mathrm{B}}(\omega))V_{\mathbf{k}\mathbf{k}'\mathbf{k}}^{ab}2i\mathrm{Im}\,\varepsilon_{\mathbf{k}-\mathbf{k}'}^{-1,\mathrm{ret}}(\omega).$$

(14)

$\varepsilon_{\mathbf{q}}^{-1,\mathrm{ret}}(\omega)$ is the longitudinal dielectric function describing screening due to excited carriers and $n^{\mathrm{B}}(\omega)$ is the Bose distribution function of the elementary plasma excitations called plasmons. $V^{ab}$ denotes Coulomb matrix elements between species $a$ and $b$ which contain dielectric screening due to carriers in the ground state and due to the environment but no screening due to excited carriers. $T''$ denotes the T-matrix with the two lowest-order terms subtracted from the ladder expansion and is discussed in the following subsection.

**T-matrix and bound carriers**. The T-matrix in statically screened ladder approximation describing bound and scattering two-particle states between carrier species $a$ and $b$ obeys a Lippmann–Schwinger equation (LSE)

$$T^{\mathrm{ret},ab}(\omega) = V^{\mathrm{S},\mathrm{ret},ab} + i\hbar V^{\mathrm{S},\mathrm{ret},ab}\mathcal{G}^{ab}(\omega)T^{\mathrm{ret},ab}(\omega),$$

(15)

where $\mathcal{G}^{\mathrm{ret},ab}(\omega)$ is the free two-particle Green's function in the particle-particle channel. The corresponding interacting two-particle Green's function $G_2^{\mathrm{ret},ab}(\omega)$ fulfills a Bethe–Salpeter equation (BSE), that has been discussed in detail in refs. [42,47] and is equivalent to the LSE. We will exploit this fact later when solving the LSE and evaluating the T-matrix self-energy. In its homogeneous form, the BSE in static ladder approximation is given by

$$0 = (E_{\mathbf{k}\sigma}^{a} + E_{\mathbf{k}+\mathbf{Q}\sigma'}^{b})G_{2,\mathbf{k},\mathbf{k}+\mathbf{Q}\sigma\sigma'}^{\mathrm{ret},ab}(\omega)$$

$$-(1 - f_{\mathbf{k}\sigma}^{a} - f_{\mathbf{k}+\mathbf{Q}\sigma'}^{b})\frac{1}{\mathcal{A}}\sum_{\mathbf{k}'}V_{\mathbf{k}+\mathbf{Q},\mathbf{k}',\mathbf{k},\mathbf{k}+\mathbf{Q}}^{\mathrm{S},ab}G_{2,\mathbf{k}',\mathbf{k}+\mathbf{Q}\sigma\sigma'}^{\mathrm{ret},ab}(\omega).$$

(16)

Diagonalization yields bound states $|\nu\sigma\sigma'\mathbf{Q}\rangle$ and eigenenergies $E_{\nu\mathbf{Q}}^{\sigma\sigma'}$. We drop the indices $a$ and $b$ here, assuming that only two-particle states between different carrier species are involved. Due to the translational invariance of the crystal, the bound states can be classified by the total exciton momentum $\mathbf{Q}$ as discussed in[48]. Here we neglect the effect of electron-hole exchange interaction that leads to a fine-

structure splitting of excitons and trions[48–50] in the meV range, which is small compared to the exciton-binding energies of several hundred meV. As a consequence, electron and hole spins, which are already good quantum numbers in monolayer TMDC materials due to crystal symmetry, also classify the bound states. For each total momentum and spin combination a series of excitons exists, which is labeled by $\nu$, analogue to the angular momentum states of Hydrogen-like Wannier excitons. Due to the two-dimensional nature of monolayer TMDCs and the related strong momentum dependence of dielectric screening, nontrivial exciton series deviating from a hydrogen-like spectrum are observed[2,51,52]. The eigenenergies approximately decompose into a part from the relative motion of electron and hole and a kinetic part depending on the total momentum: $E_{\nu\mathbf{Q}}^{\sigma\sigma'} = E_{\mathrm{rel},\nu}^{\sigma\sigma'} + E_{\mathrm{kin},\mathbf{Q}}^{\sigma\sigma'}$. We can use Bloch basis functions to find a representation of the bound states corresponding to exciton wave functions

$$\psi_{\nu\mathbf{Q}}^{\sigma\sigma'}(\mathbf{k}) = \langle\mathbf{k}\mathbf{k}'\sigma\sigma'ab|\nu\sigma\sigma'\mathbf{Q}\rangle\delta_{\mathbf{k}',\mathbf{k}+\mathbf{Q}},$$

(17)

where $\mathbf{k}$ conventionally denotes the hole momentum, while the electron momentum is fixed via the total momentum.

An explicit expression for the T-matrix can be obtained by writing the LSE (15) in the basis of two-particle eigenstates $|\nu\sigma\sigma'\mathbf{Q}\rangle$ as shown in detail in ref. [18]. Since the BSE represents a generalized eigenvalue problem, the eigenstates form a biorthogonal basis. The procedure yields a spectral representation of the T-matrix in operator form that is referred to as "bilinear expansion":

$$T^{\mathrm{ret},ab}(\omega) = \sum_{\nu\sigma\sigma'\mathbf{Q}}N_{ab}^{-1}(E_{\mathrm{kin}} - \hbar\omega)\frac{|\nu\sigma\sigma'\mathbf{Q}\rangle\langle\widetilde{\nu\sigma\sigma'}\mathbf{Q}|}{\hbar\omega - E_{\nu\mathbf{Q}}^{\sigma\sigma'}}(E_{\mathrm{kin}} - E_{\nu\mathbf{Q}}^{\sigma\sigma'})$$

(18)

with the Pauli blocking factor $N_{ab} = 1 - f^{a} - f^{b}$, the operator of kinetic energy of unbound electrons and holes $E_{\mathrm{kin}}$ and the eigenstate of the adjoint BSE $\langle\widetilde{\nu\sigma\sigma'}\mathbf{Q}|$. The bilinear expansion is used in the following to evaluate the imaginary part of the self-energy (13) and thereby the contribution of correlated carriers.

**Separation of bound and quasi-free carriers**. Inserting Eq. (13) into Eq. (11) and noting that neither the GW self-energy nor the two lowest T-matrix terms contribute to the carrier density[46], we obtain[17,19]

$$n_a(\mu_a, T) = \frac{1}{\mathcal{A}}\sum_{\mathbf{k}\sigma}f^{a}(E_{\mathbf{k}\sigma}^{a}) + \frac{1}{\mathcal{A}^2}\sum_{\mathbf{k}\mathbf{k}'b\sigma'}\int_{-\infty}^{E_{\mathrm{Gap}}}\frac{\mathrm{d}\omega}{\pi}n_{ab}^{\mathrm{B}}(\omega)$$

$$\times\,\mathrm{Im}\,T_{\mathbf{k}\mathbf{k}'\sigma\sigma'}^{\mathrm{ret},ab}(\omega)\frac{\mathrm{d}}{\mathrm{d}\omega}i\hbar\mathcal{G}_{\mathbf{k}\mathbf{k}'\sigma\sigma'}^{\mathrm{ret},ab}(\omega)$$

$$+n_{\mathrm{scatt}}.$$

(19)

$n_{ab}^{\mathrm{B}}(\omega) = [\exp(\beta(\hbar\omega - \mu_a - \mu_b)) - 1]^{-1}$ is the Bose distribution function depending on the chemical potentials of both carrier species. Equation (19) contains contributions of both bound two-particle states, which are below the single-particle gap $E_{\mathrm{Gap}}$, and scattering two-particle states. The latter are explicitly given in refs. [15–17]. Different excitons are localized at different positions in the Brillouin zone as expressed by the exciton wave functions, Eq. (17), where electrons and holes are separated by the corresponding total momentum $\mathbf{Q}$. Therefore, we do not rely on a global ($\mathbf{k}$-independent) band gap to decide whether a two-particle state is a bound state. Instead, we compare the energy of each two-particle state to the sum of electron and hole band energies at the maximum of the respective exciton wave function. The renormalization factor $Z_{\mathbf{k}\sigma}^{a}$ of the quasi-particle resonance in the spectral function (12) enters the contribution of correlated carriers as Pauli-blocking factor and as correction to the two-particle scattering spectrum. To simplify the following discussion, we neglect the contribution $n_{\mathrm{scatt}}$ of scattering states beyond the quasi-free carriers and consider only the bound-state contribution given by the real-frequency poles of the T-matrix[15]:

$$\frac{1}{\mathcal{A}^2}\sum_{\mathbf{k}\mathbf{k}'b\sigma'}\mathrm{Im}\,T_{\mathbf{k}\mathbf{k}'\sigma\sigma'}^{\mathrm{ret},ab}(\omega)\frac{\mathrm{d}}{\mathrm{d}\omega}i\hbar\mathcal{G}_{\mathbf{k}\mathbf{k}'\sigma\sigma'}^{\mathrm{ret},ab}(\omega)\bigg|_{\mathrm{bound}} = \pi\frac{1}{\mathcal{A}}\sum_{\nu\sigma'\mathbf{Q}}\delta(\hbar\omega - E_{\nu\mathbf{Q}}^{\sigma\sigma'}).$$

(20)

Using Eq. (20), we arrive at the final expression for the carrier density:

$$n_a(\mu_a, T) = \frac{1}{\mathcal{A}}\sum_{\mathbf{k}\sigma}f^{a}(E_{\mathbf{k}\sigma}^{a}) + \frac{1}{\mathcal{A}}\sum_{b\neq a}\sum_{\sigma\sigma'}\sum_{\nu\mathbf{Q}}n_{ab}^{\mathrm{B}}(E_{\nu\mathbf{Q}}^{\sigma\sigma'})$$

$$= n_{\mathrm{free}}^{\mathrm{GW},a} + n_{\mathrm{X}}.$$

(21)

The total carrier density separates into contributions from quasi-free carriers and from carriers bound as excitons according to the two poles in the spectral function $A^{a}(\omega)$. For a specific material, the ionization equilibrium has to be computed numerically. The electron and hole chemical potentials are determined by adapting the Fermi functions $f^{a}(E_{\mathbf{k}\sigma}^{a})$ of electrons and holes to a given density of quasi-free carriers at the quasi-particle energies $E_{\mathbf{k}\sigma}^{a}$. As the chemical potentials also enter the bound-carrier density via the Bose function $n_{ab}^{\mathrm{B}}$, Eq. (21) represents an implicit

equation for the fraction of quasi-free carriers $\alpha_a = n_{\text{free}}^a / n_a$, that has to be solved self-consistently with the quasi-particle energies in GW approximation, see Eq. (25), and the bound-state energies $E_{\nu\mathbf{Q}}^{\sigma\sigma'}$. To simplify the procedure, we exploit the fact that shifts of excitonic resonances are naturally much smaller than band-gap shifts, which is due to compensation effects between gap shrinkage and binding-energy reduction[33,47]. Hence, we assume that the exciton spectrum depends only weakly on the excitation density so that we can limit ourselves to the BSE (16) in the limit of zero excitation density.

Consistent with the imaginary part of the self-energy (13), the quasi-particle energies $E_{\mathbf{k}\sigma}^a$ contain GW- and T-matrix contributions:

$$E_{\mathbf{k}\sigma}^a = \varepsilon_{\mathbf{k}\sigma}^a + \text{Re}\,\Sigma_{\mathbf{k}\sigma}^{\text{GW,ret},a}(E_{\mathbf{k}\sigma}^a) + \text{Re}\,\Sigma_{\mathbf{k}\sigma}^{\text{T,ret},a}(E_{\mathbf{k}\sigma}^a). \tag{22}$$

The GW self-energy can be separated into the Fock term and the so-called Montroll–Ward term containing all contributions beyond bare exchange interaction. The T-matrix contribution is explicitly given in ref. [46] and leads to a blue shift of single-particle energies that is in the nondegenerate case ($f^a(E_{\mathbf{k}\sigma}^a) \ll 1$) caused by the bound-carrier population. At the same time, the Fock self-energy contains exchange interaction with both quasi-free and bound carriers via the extended spectral functions that leads to a lowering of single-particle energies. This can be seen by using the T-matrix self-energy in Eq. (13) to obtain an excitonic contribution to the spectral function (12) given by

$$\Gamma_{\mathbf{k}\sigma}^a(\omega) = \frac{1}{\mathcal{A}} \sum_{b \neq a} \sum_{\nu\sigma'\mathbf{Q}} \sum_{\mathbf{k}'} \left( \left| \psi_{\nu\mathbf{Q}}^{\sigma\sigma'}(\mathbf{k}) \right|^2 \delta_{a,h}\delta_{\mathbf{k}',\mathbf{k}+\mathbf{Q}} + \left| \psi_{\nu\mathbf{Q}}^{\sigma\sigma'}(\mathbf{k}-\mathbf{Q}) \right|^2 \delta_{a,e}\delta_{\mathbf{k}',\mathbf{k}-\mathbf{Q}} \right)$$

$$\times \delta\left( \hbar\omega + E_{\mathbf{k}'\sigma'}^b - E_{\nu\mathbf{Q}}^{\sigma\sigma'} \right) \left[ n_{ab}^{\text{B}}\left( E_{\nu\mathbf{Q}}^{\sigma\sigma'} \right) + f^b\left( E_{\mathbf{k}'\sigma'}^b \right) \right]. \tag{23}$$

It yields a sharp resonance for each bound state weighted by its Bose population function and the exciton wave functions at the corresponding position in $\mathbf{k}$-space. Note that the spectral positions of the resonances are not given by the bound-state energies $E_{\nu\mathbf{Q}}^{\sigma\sigma'}$, which are two-particle quantities, but by an effective binding energy of the carrier in state $|\mathbf{k}\sigma a\rangle$, as $\Gamma$ represents a single-particle spectral function. The Fock self-energy[18] can then be expressed in terms of the spectral function using the Kubo–Martin–Schwinger relation for the propagators $G^<(\omega)$ in thermal equilibrium:

$$\Sigma_{\mathbf{k}\sigma}^{F,a} = i\hbar \frac{1}{\mathcal{A}} \sum_{\mathbf{k}'} V_{\mathbf{k}\mathbf{k}'\mathbf{k}'\mathbf{k}}^{aa} G_{\mathbf{k}'\sigma}^{<,a}$$

$$= -i\hbar \frac{1}{\mathcal{A}} \sum_{\mathbf{k}'} V_{\mathbf{k}\mathbf{k}'\mathbf{k}'\mathbf{k}}^{aa} \int \frac{d\omega}{2\pi} f^a(\omega) A_{\mathbf{k}'\sigma}^a(\omega)$$

$$= -\frac{1}{\mathcal{A}} \sum_{\mathbf{k}'} V_{\mathbf{k}\mathbf{k}'\mathbf{k}'\mathbf{k}}^{aa} \left( f^a(E_{\mathbf{k}'\sigma}^a) + f_{\mathbf{k}'\sigma}^{a,\text{bound}} \right). \tag{24}$$

The first contribution to the Fock self-energy scales, besides the Coulomb matrix elements, with the free-carrier density, while the second contribution scales with the density of bound carriers. It turns out that similar to exchange interaction with free carriers, bound-carrier exchange leads to $\mathbf{k}$-dependent renormalizations according to the exciton wave functions and populations that are contained in the population factor $f_{\mathbf{k}'\sigma}^{a,\text{bound}}$. As a conclusion, the real part of the self-energy (22) contains quasi-particle renormalizations due to exciton populations via the T-matrix in two different places that act in opposite directions. We assume that these renormalizations cancel to a large degree and focus on the free-carrier contributions in accordance with refs [17,19]. Then we obtain for the quasi-particle energies:

$$E_{\mathbf{k}\sigma}^a \approx \varepsilon_{\mathbf{k}\sigma}^a + \text{Re}\,\Sigma_{\mathbf{k}\sigma}^{\text{GW,ret},a}(E_{\mathbf{k}\sigma}^a)\big|_{\text{free}}$$

$$= \varepsilon_{\mathbf{k}\sigma}^a + \Sigma_{\mathbf{k}\sigma}^{F,a}\big|_{\text{free}} + \text{Re}\,\Sigma_{\mathbf{k}\sigma}^{\text{MW,ret},a}(E_{\mathbf{k}\sigma}^a)\big|_{\text{free}} \tag{25}$$

with the Montroll–Ward contribution

$$\Sigma_{\mathbf{k}\sigma}^{\text{MW,ret},a}(\omega)\big|_{\text{free}} = i\hbar \int_{-\infty}^{\infty} \frac{d\omega'}{2\pi}$$

$$\times \sum_{\mathbf{q}} \frac{\left(1 - f^a(E_{\mathbf{q},\sigma}^a)\right) V_{\mathbf{k}\mathbf{q}\mathbf{k}\mathbf{q}}^{S,>,aa}(\omega') + f^a(E_{\mathbf{q},\sigma}^a) V_{\mathbf{k}\mathbf{q}\mathbf{k}\mathbf{q}}^{S,<,aa}(\omega')}{\hbar\omega - E_{\mathbf{q},\sigma}^a + i\gamma_{\mathbf{q},\sigma}^a - \hbar\omega'}. \tag{26}$$

The quasi-particle damping $\gamma_{\mathbf{q},\sigma}^a = -\text{Im}\,\Sigma_{\mathbf{k}\sigma}^{\text{MW,ret},a}(E_{\mathbf{k}\sigma}^a)$ is obtained from the self-consistent evaluation of the GW self-energy. It is only used for the purpose of calculating the quasi-particle energies, while the spectral function in extended quasi-particle approximation, Eq. (12), involves quasi-particle energies without broadening by construction. We assume that this is valid in a system with continuous density of states. In a similar manner as for the Fock self-energy, extended spectral functions could be used to evaluate the Montroll–Ward self-

energy in Eq. (26). Due to the spectral structure of the self-energy, however, renormalizations of the single-particle band structure caused by bound carriers involve a denominator of the order of the exciton binding energy, which is very off-resonant. Therefore the Montroll–Ward term is evaluated using spectral functions for quasi-free carriers.

**Screening due to excited carriers.** In the spirit of the extended quasi-particle approximation, dynamical screening of the Coulomb interaction due to both free carriers and bound excitons is taken into account. The free-carrier screening is treated in RPA with a macroscopic Lindhard dielectric function[18], while the excitonic polarizibilities are calculated as described in refs [53,54]:

$$\varepsilon_{\mathbf{q}}^{\text{ret}}(\omega) = \varepsilon_{\mathbf{q}}^{\text{ret,RPA}}(\omega)$$

$$- V_{\mathbf{q}} \frac{1}{\mathcal{A}} \sum_{\nu\nu'\sigma\sigma'\mathbf{Q}} \left| M_{\nu\nu'\mathbf{Q}}^{\sigma\sigma'}(\mathbf{q}) \right|^2 \frac{n_{ab}^{\text{B}}(E_{\nu\mathbf{Q}}^{\sigma\sigma'}) - n_{ab}^{\text{B}}(E_{\nu'\mathbf{Q}-\mathbf{q}}^{\sigma\sigma'})}{E_{\nu\mathbf{Q}}^{\sigma\sigma'} - E_{\nu'\mathbf{Q}-\mathbf{q}}^{\sigma\sigma'} - \hbar\omega - i\gamma} \tag{27}$$

with matrix elements

$$M_{\nu\nu'\mathbf{Q}}^{\sigma\sigma'}(\mathbf{q}) = \frac{1}{\mathcal{A}} \sum_{\mathbf{p}} \psi_{\nu\mathbf{Q}}^{\sigma\sigma'}(\mathbf{p}) \left[ \left( \psi_{\nu'\mathbf{Q}-\mathbf{q}}^{\sigma\sigma'}(\mathbf{p}+\mathbf{q}) \right)^* - \left( \psi_{\nu'\mathbf{Q}-\mathbf{q}}^{\sigma\sigma'}(\mathbf{p}) \right)^* \right] \tag{28}$$

and exciton wave functions $\psi_{\nu\mathbf{Q}}^{\sigma\sigma'}(\mathbf{p})$ as defined above. The momentum and frequency dependence of screening is characterized by the plasmon spectral function

$$\hat{V}_{\mathbf{k}\mathbf{k}'\mathbf{k}\mathbf{k}'}^{S,ab}(\omega) = V_{\mathbf{k}\mathbf{k}'\mathbf{k}\mathbf{k}'}^{S,>,ab}(\omega) - V_{\mathbf{k}\mathbf{k}'\mathbf{k}\mathbf{k}'}^{S,<,ab}(\omega)$$

$$= V_{\mathbf{k}\mathbf{k}'\mathbf{k}\mathbf{k}'}^{ab} 2i\text{Im}\,\varepsilon_{\mathbf{k}-\mathbf{k}'}^{-1,\text{ret}}(\omega). \tag{29}$$

**Fraction of bright excitons.** In the ionization equilibrium between fusion of unbound carriers and fission of excitons, all bound two-particle states with quantum number $\nu$ and total exciton momentum $\mathbf{Q}$ take part, as expressed by the exciton density, see Eq. (21). On the other hand, besides higher-order processes, only excitons with small momenta are optically active, as exciton-photon interaction requires energy and momentum conservation. From the results on the ionization equilibrium discussed in the main text, we extract the fraction of bright excitons by using the obtained electron and hole chemical potentials and summing over the appropriate exciton states in Eq. (21). To numerically resolve the exciton population function in the small window of allowed momenta, we apply an effective-mass approximation to the exciton dispersions shown in Supplementary Fig. 2 and evaluate the sum over exciton states in polar coordinates. The lowest bound-state energies involving electrons and holes with equal spins are thus given by $E_{\mathbf{Q}} \approx E_{1s} + \hbar^2 Q^2/2M$, where we find $E_{1s} = -311$ meV and $M = 1.07m_e$ for MoS$_2$ and $E_{1s} = -258$ meV and $M = 0.72m_e$ for WSe$_2$. The energies $E_{1s}$ are measured with respect to the quasi-particle band gap, while energy and momentum conservation in the exciton–photon interaction explicitly involve the band gap, restricting excitons to inside the light cone with radius $Q_{\max} = \hbar c Q_{\max} = E_{1s} + E_{\text{Gap}}$. The energy values on the right hand side correspond to the position of the A exciton in optical spectra, which we take from experiment[55] yielding 1.95 eV and 1.75 eV for MoS$_2$ and WSe$_2$ on SiO$_2$, respectively. The bright-exciton density is given by

$$n_{X,\text{bright}} = \frac{1}{\pi} \int_0^{Q_{\max}} dQ\,Q\,n_{\text{eh}}^{\text{B}}\left( E_{1s} + \frac{\hbar^2 Q^2}{2M} \right), \tag{30}$$

taking into account both $K$ and $K'$ valleys.

**Numerical details.** We calculate the ionization equilibrium from the fraction of quasi-free carriers as root of the implicit Eq. (21). The two highest valence and two lowest conduction bands are considered to cover all excitons that are relevant in a quasi-equilibrium situation. Band structures and Coulomb matrix elements are obtained from ab initio calculations as discussed in Supplementary Notes 1 and 2 and illustrated by Supplementary Fig. 1 and Supplementary Table 1. We limit the Brillouin zone to disks with radius 2.7 nm$^{-1}$ around the $K$, $K'$, $\Sigma$, $\Sigma'$ and $\Gamma$ points using a Monkhorst–Pack mesh with 30 mesh points along $\Gamma$-$M$, which yields reasonable convergence of all results. The frequency integrals involved in the Montroll–Ward self-energy (25) are extended from $-600$ to $600$ meV exploiting the relation $V_{\mathbf{k}\mathbf{k}'\mathbf{k}\mathbf{k}'}^{S,<,ab}(-\omega) = V_{\mathbf{k}\mathbf{k}'\mathbf{k}\mathbf{k}'}^{S,>,ab}(\omega)$. For simplicity, we use a dielectric function (27), which is isotropic in momentum by evaluating its dependence on $|\mathbf{q}|$ along the contour $\Gamma$-$K$ and using Coulomb matrix elements $V_{|\mathbf{q}|}$ that are averaged over Wannier orbitals, see the Supplementary Information. Both the Lindhard and the excitonic dielectric function (27) are evaluated using ground-state energies and extrapolated to the limit of vanishing phenomenological quasi-particle broadening $\gamma$. The excitonic dielectric function is evaluated for momenta $\mathbf{q}$ on the Monkhorst–Pack mesh using Eq. (27) and interpolated at arbitrary values of $|\mathbf{q}|$ using cubic Hermite splines. To reach numerical convergence of the dielectric function, we include up to 4000 bound states depending on the physical

parameters. The eigenstates and eigenvalues of the BSE, Eq. (16), are obtained by diagonalization using the SLEPc package[56] for the PETSc toolkit[57].

**Data availability**. The data that support the findings of this study are available from the corresponding author upon reasonable request.

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

## Acknowledgements

We acknowledge financial support from the Deutsche Forschungsgemeinschaft (JA 14-1 and RTG 2247 "Quantum Mechanical Materials Modelling") and the European

Graphene Flagship as well as resources for computational time at the HLRN (Hannover/Berlin). M.R. is grateful to the Alexander von Humboldt Foundation for support. We thank Michael Lorke, Christopher Gies, Paul Gartner and Dirk Semkat for fruitful discussions.

## Author contributions

A.S.: Performed the analytical and numerical calculations concerning the ionization equilibrium with support from M.F. M.R. and G.S.: Performed the ab initio calculations. All authors contributed to the interpretation of the results and to the writing of the manuscript.

## Additional information

**Competing interests:** The authors declare no competing financial interests.

