## [Peer Review File · Nature Communications]

Reviewers' comments:

Reviewer #1 (Remarks to the Author):

The paper presents a theoretical study of an ionization equilibrium between excitons and unbound electron-hole pairs in TMDC. Authors combine a powerful many-body formalism developed previously and a realistic band-structure model. They calculate signatures of equilibrium exciton concentration in the spectral function of electrons and holes and argue that they can be probed by ARPES and STM experiments.

I have found the paper to be quite interesting. The quality of research is excellent: used approximations are quite realistic; numerical calculations with the realistic band-structure model are challenging. Huge recent progress in fabrication and probing TMDC materials makes the paper very close to the present-day experiments.

As a negative part of the report, I should say that I do not see any serious progress in understanding/probing of the ionization equilibrium in electron-hole plasma. The whole theoretical formalism has been developed previously and had been used, for example, in PRB 80 155201 (2009).

To conclude, the paper partially satisfies high standards of Nature Communications. I am tending to recommend it for publishing in the journal after Authors clearly clarify several points.

1. It is stated that the population of indirect dark excitons is considerably larger than that of bright ones. I have not found any proof of that in the paper. Additional plots or estimations need to be presented. There are also two types of indirect excitons $K\sigma$ and KK' . Which of them are the most important?

2. In page 4 Authors mention that indirect dark excitons have larger binding energy than bright ones. Nevertheless, Fig.1-b presents the opposite trend: Binding energy for KK' excitons is smaller than for $K\sigma$ ones and comparable with KK' ones.

3. Authors argue that the presence of the equilibrium concentration of excitons can be deduced from ARPES and STM measurements. Semiconductors have been actively studied by these methods for decades. Have any signatures of excitons been measured previously? Why TMDC are special?

4. Recently signatures of electron-hole ionization crossover have been discussed in Coulomb drag effect in a double layer system - PRL 116, 046801 (2016). Authors may find it interesting and also discuss their results in contexts of bilayers. Electrons and holes can be probed independently there.

5. In TMDC two valleys can be probed separately by circularly polarized light. It allows selective pump of excitons and generation of their valley imbalance. Such new possibilities specific for TMDC can be also mentioned.

Reviewer #2 (Remarks to the Author):

NCOMMS-17-13988

Referee report on "Exciton fission in monolayer transition metal dichalcogenide semiconductors" by A. Steinhoff et al.

The manuscript reports the results of thorough theoretical investigation of the interplay between the excitons and the electron-hole plasma in transition metal dichalcogenide monolayers. This work is highly professional and written by experts in the field. However, I do not see sufficient motivation for publication of this work in the Nature Communications journal as compared, e.g., with the specialized journal such as 2D Materials or Physical Review B. Particularly, since "Nature Communications is an open access, multidisciplinary journal dedicated to publishing high-quality research in all areas of the biological, physical, chemical and Earth sciences. Papers published by the journal aim to represent important advances of significance to specialists within each field." (<https://www.nature.com/ncomms/about/aims>) I do not see a significance of this work for the general specialists in physics. Rather the manuscript in question is of interest for the specialists in the condensed matter physics and physics of two-dimensional materials.

Besides, I have a number of technical remarks concerning this work.

- 1) The authors put forward the theory of the ionization equilibrium between the excitons and electron-hole plasma based on the extended quasi-particle approximation and treating the self-energies within the ladder approximation. As far as I understand, the authors find substantial deviations from the Saha equation although the comparison between the simple model and the elaborate calculations is not presented.
- 2) On the level of model and approximations used it is unclear whether the approach of the authors goes beyond, e.g., Ref. [7], or the only difference is the specific form of the Coulomb interaction and the numerical values of the parameters.
- 3) The authors claim that they take into account the dynamic (frequency-dependent) screening by plasma "Beyond frequency-dependent plasma screening, we also include excitonic screening that we find to be relevant and that has not been discussed before for two-dimensional materials." (end of right column in p. 1), but I have not been able to trace this down to the calculation of the exciton binding energies. For instance, in the supplement the screening is frequency-independent (although, naturally, $\omega \rightarrow \infty$ is used due to high binding energies of excitons).
- 4) Do the authors account for the spin-orbit interaction, i.e., the spin splittings of the bands at the K-points in the Brillouin zone?
- 5) In my opinion, it would be also quite interesting to address the following issue: could it be possible due to strong Coulomb effects that both electron-hole plasma, neutral excitons as well as trions (charged excitons) may coexist simultaneously? As far as I understand the approach, it is extremely difficult to include the trions in the model of the authors, but qualitative or semiquantitative analysis would be very instructive.

In my opinion, this work is more suitable for the specialized journal and can be published after corresponding revisions and further review.

Reviewer #3 (Remarks to the Author):

The simultaneous application of ab initio DFT and methods for excited many body states is a modern development. In this context, the manuscript applies this combination to two dimensional semiconductors to determine the quasi equilibrium of correlated electron-hole pairs including a broad range of the detailed band structure: the results are important for optoelectronic applications (gain materials, doping dependence, screening dependence). The numerical procedure is probably at the leading edge of what can be done in the field, this is leading research. Even if most of the used equations are not really new, the presented material on its own is timely for the TMD field and well presented; since the manuscript aims at a quantitative understanding of TMDs (as a material class in the focus of present research), it should be published in Nature Communications. However before I can give my full recommendation I would like to ask a few question / give remarks:

-Typically, radiative damping is a process which is not comparable to the Coulomb-interaction, but

also probably not a small correction in these materials: the excitonic ground state in these materials is not a stable state cp. to, for instance the hydrogen system. How would the inclusion of these band-band / excitonic recombination (additionally electron/exciton phonon re-filling of bright states occurs) influence the results: a critical discussion and a rough estimation is recommended.

Of course, a detailed calculation of this is a different paper and is not demanded.

-On page 2: when the ARPES signal is mentioned, it is useful to provide the corresponding formula which connects to Eq.1,2.

-On page 3: it makes sense to show already Eq. 9/10 after Eq.3 –this makes the procedure more clear to an unexperienced reader, also, I think, the technical discussion how to distinguish between bound and unbound states is too late in the paper.

-Fig2 is at 300K?; that should be mentioned in the caption.

-The phenomenological Gaussian broadening seems to be strong for the lines of the spectral function, could the authors comment on the strength of the intrinsic Coulomb broadening in cp. to the additionally introduced broadening – is there a numerical reason to have such a strong phenomenological broadening?

-I, personally, think that the motivation to have a solid state counterpart of a high energy collider is pretty farfetched. In this field, there are different processes and interactions that occur: or, if I am wrong and this selling point can be substantiated with further strength it should be done! For sure, on the other hand, this is question of taste.

All in all, I can recommend the manuscript for publication with some, partly optional changes.

Reviewer #1

We thank the Reviewer for the careful examination of our manuscript and for acknowledging its excellent quality and topicality.

As a negative part of the report, I should say that I do not see any serious progress in understanding/probing of the ionization equilibrium in electron-hole plasma. The whole theoretical formalism has been developed previously and had been used, for example, in PRB 80 155201 (2009).

Response

According to this comment, we have not stated clear enough the advances of our work with respect to earlier approaches. There is serious progress in the understanding of ionization equilibrium in the sense that we combine the theory for the first time with material-realistic electronic-state calculations. These include the effects of dielectric screening due to the environment, which we identify as efficient tuning knob. The material-realistic approach also provides access to the nontrivial exciton spectrum formed by carriers from the various band-structure valleys, which is essential to understand the ionization equilibrium unlike in single-valley II-VI or III-V semiconductors. As a major conceptual development, we extend previous approaches by screening due to excitons, which is very relevant in the regime of high exciton densities as it triggers the fission of excitons into a plasma of unbound carriers. We also see great strides in the experimental verification of ionization equilibrium, as we suggest several new spectroscopic techniques to probe excitons via the spectral function. To the best of our knowledge, this point has not been made clear before by any theoretical or experimental work.

In the revised manuscript, we stress at the end of the introduction that excitonic screening has not been considered before in the discussion of ionization equilibrium. A figure (Fig. 7) is added to the Results that quantitatively demonstrates the impact of excitonic screening. We point out in the subsection on exciton satellites that ARPES and STS have not been suggested before as techniques to probe excitons.

Comment 1

It is stated that the population of indirect dark excitons is considerably larger than that of bright ones. I have not found any proof of that in the paper. Additional plots or estimations need to be presented. There are also two types of indirect excitons $K\Sigma$ and KK' . Which of them are the most important?

Response

The Reviewer has a valid point here asking for numerical evidence of our statement on dark and bright excitons. We have added a new figure (Fig. 3) and the corresponding discussion in the Results section that explain the dominance of dark excitons in the ionization equilibrium. A quantitative answer to the second question is delicate, as it requires a disentanglement of the $K\Sigma'$ exciton branch (around $Q=11 \text{ nm}^{-1}$, see Fig 1(b)) and the $K-K'$ exciton branch (below $Q=13 \text{ nm}^{-1}$). Given the fact that there are three Σ - and Σ' - valleys each, while there is only one K' -valley, we conclude that excitons involving Σ and Σ' are more important.

Comment 2

In page 4 Authors mention that indirect dark excitons have larger binding energy than bright ones. Nevertheless, Fig.1-b presents the opposite trend: Binding energy for KK excitons is smaller than for K\Sigma ones and comparable with KK' ones.

Response

We do not see a contradiction here: It is claimed in the manuscript that K-Sigma excitons have larger binding energies than K-K excitons, which we underline with an additional reference in the revised manuscript. This result is reflected in Fig. 1(b), where K-Sigma excitons are below the upper branch of K-K excitons belonging to the upper conduction band and, hence, to equal electron and hole spins. The K-K excitons with different electron and hole spins are even lower due to the large spin-orbit splitting in WS₂ and overcompensates the larger binding energy of K-Sigma. For better visibility, we have added parabolic fits to the K-K exciton dispersion as guide to the eye in Fig. 1(b).

Comment 3

Authors argue that the presence of the equilibrium concentration of excitons can be deduced from ARPES and STM measurements. Semiconductors have been actively studied by these methods for decades. Have any signatures of excitons been measured previously? Why TMDC are special?

Response

We are not aware of any reports on the observation of excitons by means of photoemission or tunneling spectroscopy. Such an experiment requires optical excitation of the semiconductor before probing to generate an observable population of excitons, which demands time-resolved spectroscopy. This is a rather recent development for ARPES, where time and energy resolution have improved significantly in recent years. It is also advantageous to have a spectral separation of excitons and quasi-particle bands that is well above the energy resolution of the method, which makes two-dimensional semiconductors particularly interesting to perform this kind of experiments. Moreover, two-dimensional materials have proven to be very sensitive to ARPES as stated for example in [PRB 95, 041405 (2017)]. We make a clearer statement about experiments in the revised manuscript.

Comment 4

Recently signatures of electron-hole ionization crossover have been discussed in Coulomb drag effect in a double layer system - PRL 116, 046801 (2016). Authors may find it interesting and also discuss their results in contexts of bilayers. Electrons and holes can be probed independently there.

Response

We thank the Reviewer for drawing our attention to the literature on the exciton-plasma balance in bilayer systems, which widens the perspective of our paper. We have added a discussion on excitons in TMDC bilayers in the revised manuscript.

Comment 5

In TMDC two valleys can be probed separately by circularly polarized light. It allows selective pump of excitons and generation of their valley imbalance. Such new possibilities specific for TMDC can be also mentioned.

Response

We have added a comment regarding valley-selective excitation of excitons in the revised manuscript.

Reviewer #2

We thank the Reviewer for the detailed review and we are pleased by the assessment of our work as highly professional.

„However, I do not see sufficient motivation for publication of this work in the Nature Communications journal as compared, e.g., with the specialized journal such as 2D Materials or Physical Review B. Particularly, since “Nature Communications is an open access, multidisciplinary journal dedicated to publishing high-quality research in all areas of the biological, physical, chemical and Earth sciences. Papers published by the journal aim to represent important advances of significance to specialists within each field.” (<https://www.nature.com/ncomms/about/aims>) I do not see a significance of this work for the general specialists in physics. Rather the manuscript in question is of interest for the specialists in the condensed matter physics and physics of two-dimensional materials.“

Response

We would like to emphasize that this point of view is not shared by the other Reviewers. Moreover, condensed matter physics is a very wide field by itself, i.e., interest to the condensed matter physics community already means very broad interest. We even address physicists beyond this field, pointing out the analogy of exciton fission to high-energy collider experiments.

Comment 1

The authors put forward the theory of the ionization equilibrium between the excitons and electron-hole plasma based on the extended quasi-particle approximation and treating the self-energies within the ladder approximation. As far as I understand, the authors find substantial deviations from the Saha equation although the comparison between the simple model and the elaborate calculations is not presented.

Response

It is not intended in the paper to directly compare the full solution for the ionization equilibrium with results from Saha's equations, in particular because the assumption of a single band-structure valley is not appropriate for TMDCs. As we state in the manuscript (before the old Eq. (6)), all numerical calculations are performed using the full theory, while Saha's equation is introduced only

for the purpose of illustration.

Comment 2

On the level of model and approximations used it is unclear whether the approach of the authors goes beyond, e.g., Ref. [7], or the only difference is the specific form of the Coulomb interaction and the numerical values of the parameters.

Response

This is a very helpful remark, as it shows that we did not point out the theoretical advances of our work clear enough. As a major conceptual development, we extend prior approaches by screening due to excitons, which is very relevant in the regime of high exciton densities as it triggers the fission of excitons into a plasma of unbound carriers. In the revised manuscript, we stress at the end of the introduction that excitonic screening has not been considered before in the discussion of ionization equilibrium. A figure (Fig. 7) is added to the Results that quantitatively demonstrates the impact of excitonic screening. Moreover, by combining advanced ab initio modelling techniques (GW,RPA), newly developed multi-scale theories (WFCE) and the methods from Ref. 7, our work is an important step towards material-realistic modelling of complex phase diagrams in excited 2-d materials and heterostructures thereof.

Comment 3

The authors claim that they take into account the dynamic (frequency-dependent) screening by plasma “Beyond frequency-dependent plasma screening, we also include excitonic screening that we find to be relevant and that has not been discussed before for two-dimensional materials.” (end of right column in p. 1), but I have not been able to trace this down to the calculation of the exciton binding energies. For instance, in the supplement the screening is frequency-independent (although, naturally, $\omega = \infty$ is used due to high binding energies of excitons).

Response

The Reviewer is right, we did not take into account dynamical screening effects in the calculation of exciton energies. One reason for using a static approximation to the T matrix is a purely practical one, since a dynamically screened T matrix depends on three instead of one frequency arguments, which makes the whole approach numerically unsolvable. On the other hand, exciton eigenenergies show a much weaker dependence on excitation density than single-particle energies due to compensation effects of various many-body contributions. This justifies the used approximations to the T matrix, while single-particle renormalizations should be treated on a more elaborate level, which we do by including plasma and excitonic screening there.

Comment 4

Do the authors account for the spin-orbit interaction, i.e., the spin splittings of the bands at the K-points in the Brillouin zone?

Response

Yes, we do account for spin-orbit interaction. To clarify this to the reader, we have added a statement to the legend of Fig.1(a), where an exemplary band structure is shown.

Comment 5

In my opinion, it would be also quite interesting to address the following issue: could it be possible due to strong Coulomb effects that both electron-hole plasma, neutral excitons as well as trions (charged excitons) may coexist simultaneously? As far as I understand the approach, it is extremely difficult to include the trions in the model of the authors, but qualitative or semiquantitative analysis would be very instructive.

Response

We agree that this is a very interesting question. Qualitatively, we would expect that coexistence of all three phases is possible at room temperature, as some of the exciton population will be drawn to bound-trion states. However, any numerical evaluation of this by including three-particle complexes as new species into the ionization equilibrium is unfortunately out of reach at the moment. In order to nevertheless address this point, we have added a qualitative statement to the manuscript.

Reviewer #3

We are very grateful for the decidedly positive assessment of our work and for the appreciation of its timeliness and quality of presentation.

Comment 1

Typically, radiative damping is a process which is not comparable to the Coulomb-interaction, but also probably not a small correction in these materials: the excitonic ground state in these materials is not a stable state cp. to, for instance the hydrogen system. How would the inclusion of these band-band / excitonic recombination (additionally electron/exciton phonon re-filling of bright states occurs) influence the results: a critical discussion and a rough estimation is recommended. Of course, a detailed calculation of this is a different paper and is not demanded.

Response

The Reviewer has a valid point here. The recombination of electrons and holes is an important effect in TMDC semiconductors and demands a critical discussion, which we have added in the revised manuscript. According to experimental and theoretical results, the relaxation and equilibration of excitons and quasi-free carriers are faster than recombination. As a result, empty states are immediately refilled and the system practically remains in quasi-equilibrium, where the exciton-plasma balance adiabatically adapts to the time-dependent excitation density. This picture is consistent with the discussion of experimental results in Ref. 22.

Comment 2

On page 2: when the ARPES signal is mentioned, it is useful to provide the corresponding formula which connects to Eq.1,2.

Response

The corresponding formula has been added in the revised manuscript as Eq. (3).

Comment 3

On page 3: it makes sense to show already Eq. 9/10 after Eq.3 –this makes the procedure more clear to an unexperienced reader, also, I think, the technical discussion how to distinguish between bound and unbound states is too late in the paper.

Response

We see the point that the right balance between qualitative discussion and technical details has to be found to guarantee good accessibility for readers with different theoretical experience. However, we do not see the need to show Eq. (9) in the main text, as the carrier density is already defined there. Our suggestion is to show Eq. (10) before the old Eq. (3) to clarify the separation of carriers into excitons and plasma via the spectral function. Moreover, we have added a comment about the technical separation between the species in the main text and have moved the description of the numerical procedure of separation forward in the Methods section.

Comment 4

Fig2 is at 300K?; that should be mentioned in the caption.

Response

Indeed the temperature is 300 K, we have added a remark in the caption.

Comment 5

The phenomenological Gaussian broadening seems to be strong for the lines of the spectral function, could the authors comment on the strength of the intrinsic Coulomb broadening in cp. to the additionally introduced broadening – is there a numerical reason to have such as strong phenomenological broadening?

Response

The phenomenological broadening is chosen such that a good presentation of the single-particle spectral functions is enabled while being of reasonable magnitude. The approach to ionization equilibrium via the extended quasi-particle approximation involves only quasi-particle energies (real part of the self-energy) but no quasi-particle lifetimes (imaginary part of the self-energy) in the spectral functions. However, quasi-particle lifetimes are obtained at the same level as quasi-particle energies from the self-consistent evaluation of the complex GW self-energy. Here we find lifetimes on the order of 10 meV. We believe that quasi-particle lifetimes should be subject to a different paper, although this question from the Reviewer motivates us to be more precise in the discussion of the GW self-energy in the revised manuscript.

Comment 6

I, personally, think that the motivation to have a solid state counterpart of a high energy collider is pretty farfetched. In this field, there are different processes and interactions that occur: or, if I am wrong and this selling point can be substantiated with further strength it should be done! For sure, on the other hand, this is question of taste.

Response

We agree that in collider experiments nuclear forces instead of Coulomb forces have to be overcome to fissure compound particles. However, the analogy we suggest is independent of the

nature of binding energy that is involved. In our opinion, the analogy between exciton fission in ARPES or STS and collider experiments on composite bosons is quite appealing and might be interesting to many readers. We have strengthened this point by giving concrete examples of collider experiments in the discussion.

REVIEWERS' COMMENTS:

Reviewer #1 (Remarks to the Author):

I have found the response to my questions to be convincing and satisfactory. All raised points are reflected. Moreover, Authors have performed additional calculations and included the corresponding results to the new version of the paper.

To conclude, the paper partially satisfies high standards of Nature Communications. It is a little bit too technical for a general reader, as it also has been noted by Referee #2. Nevertheless, the quality of research is considerably higher than that of a typical Physical Review paper. Predictions are experimentally testable and could motivate further experimental study of TMDC. That is why I am tending to recommend the paper to be published in Nature Communications.

Reviewer #2 (Remarks to the Author):

NCOMMS-17-13988

Second report on "Exciton fission in monolayer transition metal dichalcogenide semiconductors" by A. Steinhoff et al.

In the revised version of the manuscript the authors have addressed my main technical comments as well as the technical comments made by other referees. However, I am still not convinced that this manuscript is suitable for Nature Communications. I do not find the response of the authors to this point ("We would like to emphasize that this point of view is not shared by the other Reviewers. Moreover, condensed matter physics is a very wide field by itself, i.e., interest to the condensed matter physics community already means very broad interest. We even address physicists beyond this field, pointing out the analogy of exciton fission to high-energy collider experiments.") satisfactory. Moreover, referee 1 raised a similar question "As a negative part of the report, I should say that I do not see any serious progress in understanding/probing of the ionization equilibrium in electron-hole plasma. The whole theoretical formalism has been developed previously and had been used, for example, in PRB 80 155201 (2009)." and the authors merely reformulated a bit the end of the introduction including the point on the screening by the excitons.

So far, I cannot recommend the publication of this work in Nature Communications, although, I am most sure that this work is fully suitable for specialized journal like Physical Review B or 2D Materials.

Reviewer #1

I have found the response to my questions to be convincing and satisfactory. All raised points are reflected. Moreover, Authors have performed additional calculations and included the corresponding results to the new version of the paper.

To conclude, the paper partially satisfies high standards of Nature Communications. It is a little bit too technical for a general reader, as it also has been noted by Referee #2. Nevertheless, the quality of research is considerably higher than that of a typical Physical Review paper. Predictions are experimentally testable and could motivate further experimental study of TMDC. That is why I am tending to recommend the paper to be published in Nature Communications.

Response

We are very pleased that the Reviewer found our response convincing and again acknowledges the high quality of our research, thereby recommending publication in Nature Communications. We see the point that our paper might be a bit more technical than other Nature Communication papers. Nevertheless we feel that the main text and its key results are accessible without referring to the technical details given in the Methods section as we provide a simple chemical picture to illustrate the underlying physical processes.

Reviewer #2

In the revised version of the manuscript the authors have addressed my main technical comments as well as the technical comments made by other referees. However, I am still not convinced that this manuscript is suitable for Nature Communications. I do not find the response of the authors to this point (“We would like to emphasize that this point of view is not shared by the other Reviewers. Moreover, condensed matter physics is a very wide field by itself, i.e., interest to the condensed matter physics community already means very broad interest. We even address physicists beyond this field, pointing out the analogy of exciton fission to high-energy collider experiments.”) satisfactory. Moreover, referee 1 raised a similar question “As a negative part of the report, I should say that I do not see any serious progress in understanding/probing of the ionization equilibrium in electron-hole plasma. The whole theoretical formalism has been developed previously and had been used, for example, in PRB 80 155201 (2009).” and the authors merely reformulated a bit the end of the introduction including the point on the screening by the excitons.

So far, I cannot recommend the publication of this work in Nature Communications, although, I am most sure that this work is fully suitable for specialized journal like Physical Review B or 2D Materials.

Response

We are sorry to hear that the Referee is still not convinced of the suitability of our paper for Nature Communications. We did not only reformulate the introduction but also provided additional results that underline the importance of screening by excitons and therefore demonstrate the significant development of our theory beyond the theory used in previous papers. We would like to stress that Referee 1 acknowledges these additional results in our revised manuscript and considers this critical point as being reflected satisfactorily.